# Effects of Uncertainty, Appraisal of Uncertainty, and Self-Efficacy on the Quality of Life of Elderly Patients with Lung Cancer Receiving Chemotherapy: Based on Mishel’s Theory of Uncertainty

**DOI:** 10.3390/medicina59061051

**Published:** 2023-05-30

**Authors:** Min-Kyung Hwang, Hee-Kyung Kim, Ki-Hyeong Lee

**Affiliations:** 1Department of Nursing, Graduate School, Kongju National University, Gongju 32588, Republic of Korea; skyhmk0024@naver.com; 2Department of Nursing, Kongju National University, Gongju 32588, Republic of Korea; 3Division of Medical Oncology, Chungbuk National University Hospital, Cheongju 28644, Republic of Korea; kihee@chungbuk.ac.kr

**Keywords:** elderly, lung cancer, chemotherapy, uncertainty, appraisal of uncertainty, self-efficacy, quality of life

## Abstract

*Background and Objectives:* The purpose of this study is to enhance the quality of life in elderly patients with lung cancer by understanding relations of uncertainty, appraisal of uncertainty, self-efficacy, and quality of life targeting elderly patients with lung cancer receiving anticancer therapy, and also analyzing the factors affecting the quality of life based on Mishel’s theory. *Materials and Methods:* The subjects were a total of 112 lung cancer patients aged 65 or older receiving anticancer therapy. The data was collected by using self-report questionnaires targeting patients in hemato-oncology at Chungbuk National University Hospital. The data were analyzed using descriptive statistics, a *t*-test, an analysis of variance, Pearson’s correlational coefficients, and hierarchical regression analysis. *Results:* In stage 1, anticancer therapy (chemotherapy) (β = −0.34, *p* < 0.001), economic condition (low) (β = −0.30, *p* < 0.001), the number of anticancer therapies (three times or more) (β = −0.29, *p* < 0.001), and education (graduation from high school or higher) (β = 0.18, *p* = 0.033) were influencing factors (F = 0.52, *p* < 0.001). In stage 2, self-efficacy (β = 0.41, *p* < 0.001), appraisal of uncertainty: danger (β = −0.29, *p* < 0.001), appraisal of uncertainty: opportunity (β = 0.18, *p* = 0.018), the number of anticancer therapies (three times or more) (β = −0.17, *p* = 0.006), and anticancer therapy (chemotherapy) (β = −0.14, *p* = 0.031) were influencing factors, which showed 74.2% explanatory power (F = 26.17, *p* < 0.001). *Conclusions:* In order to improve the quality of life of subjects, it would be necessary to develop interventions for raising their self-efficacy by considering their degree of education, economic condition, the types and numbers of anticancer therapies, and understanding of the appraisal of uncertainty about the disease is assessed as an opportunity factor or a danger factor.

## 1. Introduction

In Republic of Korea, the elderly population aged 65 or older was 14.9% in 2019 [1]. Furthermore, the incidence rate of cancer in people aged 65 or older was 47.8% of all patients, mainly due to the natural increase in cancer incidence following an aging population, making it a significant cause for the recently increased number of cancer patients [2].

Meanwhile, thanks to the recent expansion of medical examinations and the development of treatment methods, the five-year relative survival rate of lung cancer patients has also improved [1]. This means that the period of struggling against cancer is extended after diagnosis [3]. Furthermore, with the increased survival rate of cancer patients, the medical team’s interest is gradually expanding from reducing tumor size to improving the quality of life of patients [4]. In the case of lung cancer patients, the question of “what kind of life he/she should live?” is becoming increasingly important [5], making the quality of life a treatment goal second only to the survival rate.

In fact, recent studies have revealed that quality of life is a strong predictive factor of survival, highlighting its growing importance [6,7]. For instance, in a study conducted by Yun et al. [8], the risk of death for lung cancer patients with reduced health-related quality of life after surgery was more than twice as high as that of patients with no change in quality of life. This underscores the importance of managing and improving the quality of life of lung cancer patients.

Moreover, the symptoms that lung cancer patients experience can have a negative impact on their physical, psychological, social, and spiritual well-being [9]. Inadequate management of these symptoms may cause a decline in quality of life [10,11]. Therefore, it is crucial to properly manage the symptoms and improve the quality of life of lung cancer patients.

The quality of life is a state of well-being that is subjectively appraised based on the general and overall situation or life experience in multidimensional areas such as physical, psychological, social, and functional [12]. For cancer patients, the quality of life is influenced by factors such as clinical stage, age, and performance status at the time of diagnosis [13]. In addition, greater uncertainty about their condition can lead to a decline in their overall quality of life [14,15,16]. In another study by Kim [17], comparing uncertainty levels between adults and elderly patients with lung cancer, the elderly had higher levels of uncertainty about the disease due to sociodemographic and disease-related characteristics.

Uncertainty refers to a lack of understanding of one’s disease, related treatments, post-treatment prognosis, and the inability to judge disease-related situations. If uncertainty persists throughout the course of the disease’s progression, it can be considered threatening as it delays the formation of cognitive structures and limits the ability to properly assess an individual’s situation [18]. According to Mishel [19], uncertainty in disease-related situations arises when decision-makers cannot accurately evaluate goals or predict results. Therefore, it is important for experts to help reduce uncertainty by assisting patients in recognizing disease conditions and situations. Furthermore, Mishel [20] notes that individuals in a state of chronic uncertainty shift from trying to avoid it to accepting it as a new perspective on life and a part of reality.

Since uncertainty in the early stages of disease becomes a changeable and destructive element, a patient appraises it as a ‘danger’. Once the uncertainty is continued, a certain order is built, and the uncertainty is appraised positively. In this case, it could be regarded as an ‘opportunity’ in life [21], so the appraisal of uncertainty is also important. On the other hand, if it is appraised as an opportunity, the ability to cope with the disease is improved, and a new view of life could be established [22]. Thus, in order for patients to be able to positively cope with the disease process, it is important to help them integrate uncertainty that was appraised dangerous as a part of their lives and change into a positive opportunity [23].

In addition, self-efficacy is a dynamic process of appraising one’s own ability to perform actions necessary for coping with and adjusting to a potentially-threatening event [24]. Due to the expectation that a certain act brings about a certain result and confidence in the successful performance of a certain act having huge effects on actions [24], cancer patients can continuously maintain their disease control by having self-efficacy [25]. Self-efficacy, which is confidence in managing chronic disease, is an important factor for enjoying a healthy life and enhancing the quality of life [26]. For this reason, it plays an important role in cancer patients’ self-management [27,28]. In a study conducted by Ko [29] on patients with non-small cell lung cancer, higher levels of self-efficacy were found to be associated with a better quality of life. For cancer patients, quality of life is as important a goal as survival, so there has been much research on it, but it is hard to find research targeting elderly patients with lung cancer. Therefore, the purpose of this study is to evaluate the relations of uncertainty, appraisal of uncertainty: danger and opportunity, self-efficacy, and quality of life targeting elderly patients with lung cancer receiving anticancer therapy, and also analyze the factors affecting the quality of life based on Mishel’s theory of uncertainty.

## 2. Materials and Methods

### 2.1. Research Design and Subjects

This study is a descriptive survey research study for understanding the relations between uncertainty, appraisal of uncertainty, self-efficacy, and quality of life targeting elderly patients with lung cancer and also analyzing the factors affecting the quality of life. The research subjects were 112 elderly patients with lung cancer receiving anticancer therapy from a tertiary hospital in Chungbuk. The criteria for selection are as follows: (1) male and female senior citizens 65 or older; (2) people who received chemotherapy once or more after getting diagnosed with primary lung cancer; (3) people who have insight into the disease and an ability to communicate; and (4) people who understand the purpose of this study and have submitted written consent to participate in this study. The criteria for exclusion are people who were diagnosed with a cognitive disorder or mental illness, which was verified in their medical records. The researcher directly distributed 114 questionnaires and then conducted a face-to-face survey. After excluding two improper questionnaires, 112 people were finally included as research subjects.

### 2.2. Research Instruments

#### 2.2.1. General Characteristics and Disease-Related Characteristics

This study verified sex, age, marital status, main caregiver, degree of education, breadwinner, and economic condition as the sociodemographic characteristics and also researched diagnosis, clinical stage of lung cancer, types of anticancer therapy, and the number of anticancer therapies as the disease-related characteristics.

#### 2.2.2. Uncertainty

In order to measure uncertainty, this study used the instrument translated into Korean by Chung et al. [30] in 2005 based on Mishel’s uncertainty in illness scale-community form (MUIS-C) developed by Mishel [19]. This instrument is composed of four sub-concepts and 33 items, such as seven items about discordance between diagnosis and seriousness of the disease, thirteen items about the ambiguity of disease, seven items about the complexity of the nursing system and treatment, five items about the unpredictability of disease and prognosis, and one item that did not belong to any of them. Each item is based on the 5-point Likert scale (1 point for ‘Not at all’, 5 points for ‘Very much so’). The mean ratings are 1–5 points. A higher score means a high degree of uncertainty. When the instrument was initially developed, the reliability was shown as Cronbach’s α = 0.91~0.93. In the research by Chung et al. [30] the Cronbach’s α was 0.85. In this study, Cronbach’s α = 0.87.

#### 2.2.3. Appraisal of Uncertainty

For the appraisal of uncertainty, this study used the instrument translated and verified for validity by Kang [31] based on the appraisal scale developed by Mishel and Sorenson [21]. The uncertainty appraisal scale is composed of two sub-areas and 15 items, such as eight items about danger appraisal and seven items about opportunity appraisal. Each item is based on the 6-point Likert scale (0 points for ’Not at all’, 5 points for ‘Very much so’). The mean ratings are 0–5 points. In the danger appraisal, when the score is higher, uncertainty is appraised as danger. In the opportunity appraisal, when the score is higher, uncertainty is appraised as an opportunity.

In the research by Mishel and Sorenson [21], the reliability of appraisal of uncertainty: danger was shown as Cronbach’s α = 0.87, and the reliability of appraisal of uncertainty: the opportunity was shown as Cronbach’s α = 0.82. In this study, the reliability of appraisal of uncertainty: danger was shown as Cronbach’s α = 0.88, and the reliability of appraisal of uncertainty: the opportunity was shown as Cronbach’s α = 0.91.

#### 2.2.4. Self-Efficacy

Self-efficacy was measured using the Korean-version instrument (Cancer Survivors’ Self-Efficacy Scale–Korean, CSSES-K) [32] of the cancer survivors’ self-efficacy scale developed by Foster et al. [33]. CSSES-K is composed of a total of ten items, including five items about the management of health problems and five items about the pursuit of help and support. The mean ratings of each item are 1–10 points, from 1 point for ‘Not confident at all’ to 10 points for ‘Very confident’. A higher score means high self-efficacy. When the instrument was initially developed, the reliability was shown as Cronbach’s α = 0.92, whereas it is α = 0.89 in this study.

#### 2.2.5. Quality of Life

The quality of life was measured through the Korean-version functional assessment of cancer therapy scale-lung (FACT-L), version 4 [34], developed by Functional Assessment of Chronic Illness Therapy (FACIT). It is composed of a total of five sub-areas and 36 items, including seven items about the body area, six items about the society/family area, six items about the emotion area, seven items about the function area, and ten items about the disease characteristics area. Each item is based on the 5-point Likert scale (0 points for ‘Not at all’, 4 points for ‘Very much so’). It was evaluated through 33 items except for three unscored items in the disease characteristics area (lung cancer-related factor). The mean ratings are 0–4 points. A higher score means a high health-related quality of life. In the research by Yoo et al. [35], the reliability of the Korean-version instrument was shown as Cronbach’s α = 0.86. In this study, the reliability was shown as Cronbach’s α = 0.92.

### 2.3. Data Collection and Ethical Considerations

The data were collected from 12 October 2022 to 27 January 2023 after obtaining approval from the institutional review board of Chungbuk National University Hospital. The research subjects were elderly patients with lung cancer suitable for the criteria of selection among outpatients and hospitalized patients in hemato-oncology. I met the subjects in the outpatient counseling room of the Department of Hematology at Chungbuk National University Hospital and collected data. The researcher fully explained the purpose and methods of this study, the confidentiality related to participation in this study, and the possibility of withdrawing their participation at any time with no damage to the subjects. After that, the patients voluntarily signed the written consent for their participation. The subjects filled out the self-report questionnaires for themselves. For the patients who found it difficult to fill out the questionnaire, the researcher directly read the content of the questionnaire and took down their responses. It took an average of 20–30 min to complete the survey. The collected data was kept in a cabinet with locks. A password was set up for the PC, including the research data, so no one else but the researcher could read the data.

### 2.4. Data Analysis

Using the SPSS^®^ Statistics 27.0 software made by IBM Corporation, Armonk, NY, USA, the general characteristics and disease-related characteristics of subjects were calculated through descriptive statistics such as real numbers, percentages, mean, and standard deviation. This study conducted the *t*-test and ANOVA for comparing the quality of life according to the general characteristics and disease-related characteristics of subjects and used the Scheffe test for the post-analysis. This study used the mean and standard deviation for the degree of uncertainty, appraisal of uncertainty: opportunity, appraisal of uncertainty: danger, self-efficacy, and quality of life of subjects, used Pearson’s correlational coefficients for relations of uncertainty, appraisal of uncertainty: opportunity, appraisal of uncertainty: danger, self-efficacy, and quality of life of subjects, and conducted hierarchical regression analysis for the factors affecting the quality of life of subjects.

## 3. Results

### 3.1. General Characteristics and Disease-Related Characteristics of Participants

The general characteristics and disease-related characteristics of subjects are listed in Table 1. In terms of general characteristics, the number of elderly patients with lung cancer was 112. The average age was 70.98 *±* 5.65 years, and the range of age was 65~87 years. The most responses were obtained in married couples (*n* = 79, 70.5%) for marital status, spouses (n = 62, 55.4%) for main caregivers, and oneself or pensioners (*n* = 68, 60.7%) for breadwinners. In terms of disease-related characteristics, most responses were obtained in non-small cell lung cancer (*n* = 103, 92.0%) for diagnosis and stage III or IV (*n* = 103, 92.0%) for the clinical stage. The types of anticancer therapy included immunotherapy (*n* = 37, 33.0%), targeted therapy (*n* = 35, 31.3%), chemotherapy (*n* = 28, 25.0%), and radiotherapy and chemotherapy (*n* = 12, 10.7%). In terms of the number of people receiving anticancer therapy, 73 people (65.2%) received it once.

### 3.2. Differences in Health-Related Quality of Life According to General Characteristics and Disease-Related Characteristics

The comparison of the quality of life according to the general characteristics and disease-related characteristics of subjects is shown in Table 1. There were differences in the quality of life according to the degree of education (t = −2.48, *p* = 0.015), economic condition (t = 3.69, *p* < 0.001), types of anticancer therapy (F = 5.94, *p* = 0.001), and number of anticancer therapies (F = 5.88, *p* = 0.004) of subjects. In other words, people who graduated from high school or higher showed a higher quality of life than people who graduated from middle school or lower. The people who responded to economic conditions as middle showed a higher quality of life than the people who responded as low. In terms of the types of anticancer therapy, people who received immunotherapy or targeted therapy showed a higher quality of life than people who received chemotherapy. In the number of receiving anticancer therapy, the people with the first or second time showed higher quality of life than people with three times or more. There were no differences in the quality of life according to other general characteristics or disease-related characteristics.

### 3.3. Degree of Uncertainty, Appraisal of Uncertainty: Opportunity, Appraisal of Uncertainty: Danger, Self-Efficacy, and Quality of Life of Participants

The degree of uncertainty, appraisal of uncertainty: danger, appraisal of uncertainty: opportunity, self-efficacy, quality of life, and sub-areas of quality of life are shown in Table 2. The degree of uncertainty was 2.62 ± 0.49 out of 4 points. The degree of appraisal of uncertainty: danger was 1.49 ± 0.99 out of 5 points, and the degree of appraisal of uncertainty: the opportunity was 2.36 ± 1.03 out of 5 points. The degree of self-efficacy was 6.04 ± 1.53 out of 10 points, while the degree of quality of life was 2.58 ± 0.95 out of 4 points. In the subareas of the quality of life, the physical state was 2.95 ± 0.80 points: the social/family state was 2.46 ± 0.91 points; the emotional state was 2.82 ± 0.82 points; the functional state was 2.02 ± 0.89 points; and the other state was 2.65 ± 0.75 points.

### 3.4. Correlations of Uncertainty, Appraisal of Uncertainty: Opportunity, Appraisal of Uncertainty: Danger, Self-Efficacy, and Quality of Life

The relations of uncertainty, appraisal of uncertainty: danger, appraisal of uncertainty: opportunity, self-efficacy, and quality of life of subjects are shown in Table 3. The quality of life of elderly patients with lung cancer showed moderate correlations with uncertainty (r = −0.56, *p* < 0.001), appraisal of uncertainty: danger (r = −0.61, *p* < 0.001), appraisal of uncertainty: opportunity (r = 0.58, *p* < 0.001), and self-efficacy (r = 0.70, *p* < 0.001). In other words, the higher the degree of uncertainty and the higher the appraisal of uncertainty as a danger, the lower the quality of life of the participants. In addition, the higher the degree of self-efficacy and the higher the appraisal of uncertainty as an opportunity, the higher the quality of life of the participants.

### 3.5. Factors Affecting the Quality of Life of Elderly Patients with Lung Cancer

The analysis on the factors affecting the quality of life of subjects are presented in Table 4. For the factors affecting the quality of life of subjects, this study conducted a multiple regression analysis by changing the degree of education, economic condition, types of anticancer therapy, and the number of anticancer therapies verified as showing significant differences from the quality of life among the general characteristics as dummy variables and then putting them, as well as uncertainty, appraisal of uncertainty as an opportunity factor, appraisal of uncertainty as a danger factor, and self-efficacy shown to have correlations with the quality of life. In the results of testing multicollinearity for the multiple regression analysis, the tolerance limit was 0.47–0.77, which was higher than 0.1. The variance inflation factor (VIF) was 1.30–2.13, which was lower than 10, so there were no problems with multicollinearity. In the results of the residual analysis through the Durbin–Watson test, it was −2.08~2.40, close to 2, so the homogeneity of variances was validated. There was no autocorrelation between error terms in the model. In the results of the residual analysis, all the linearity, normality, and homogeneity of the model were satisfied. 

In the results of conducting the hierarchical regression analysis, in step 1, the degree of education, economic condition, types of anticancer therapy, and the number of anticancer therapies that showed significant differences among the general characteristics and disease-related characteristics were input. As a result, the variables such as anticancer therapies (chemotherapy) (β = −0.34, *p* < 0.001), economic condition (low) (β = −0.30, *p* < 0.001), the number of anticancer therapies (three times or more) (β = −0.29, *p* < 0.001), and graduation from high school or higher (β = 0.18, *p* = 0.033) showed statistically significant results, and those variables explained 36.5% of the quality of life of subjects (F = 8.52, *p* < 0.001). In the results of analyzing by adding uncertainty, appraisal of uncertainty: danger, appraisal of uncertainty: opportunity, and self-efficacy to the general characteristics and disease-related characteristics in step 2, self-efficacy (β = 0.41, *p* < 0.001), appraisal of uncertainty: danger (β = −0.29, *p* < 0.001), appraisal of uncertainty: opportunity (β = 0.18, *p* = 0.018), the number of anticancer therapy (three times or more) (β = −0.17, *p* = 0.006), anticancer therapy (chemotherapy) (β = −0.14, *p* = 0.031) were significant variables explaining the quality of life. As the explanatory power was increased by 37.7% compared to step 1, it totaled 74.2%, which was statistically significant (F = 26.17, *p* < 0.001).

In other words, to increase the quality of life of elderly patients with lung cancer receiving anticancer therapy, the types and number of anticancer therapies would need to be considered. In addition, when the research subjects appraised uncertainty as an opportunity when self-efficacy was raised more when the quality of life was higher, and when the degree of appraising uncertainty as the danger was lower, the quality of life got higher. Thus, variables such as an appraisal of uncertainty: opportunity, appraisal of uncertainty: danger and self-efficacy should be considered for elderly patients with lung cancer. 

## 4. Discussion

This study aims to understand the relations between uncertainty, appraisal of uncertainty, self-efficacy, and quality of life targeting elderly patients with lung cancer receiving anticancer therapy, to analyze the factors affecting the quality of life, and to discuss the main results. 

In this study, uncertainty averaged 2.62 points. It was 2.61 points in the research by Mo [36] targeting lung cancer patients and 2.66 points in the research by Jeong [14], which was similar to and supported the results of this study. This score is a bit higher than the uncertainty of patients with other types of cancer [23]. As lung cancer is hard to completely heal, highly recurrent, and accompanied by severe somatic symptoms such as difficulty breathing and high psychological discomfort, the prognosis is unclear in the future, so patients highly experience uncertainty [15]. Especially elderly patients with lung cancer cannot clearly perceive the state of the disease due to the decline in psychological, physical, and cognitive coping abilities, which is why uncertainty is more experienced. Thus, uncertainty should be reduced by providing information about the state of the disease and treatment. 

In the appraisal of uncertainty in subjects, the degree of appraising uncertainty as an opportunity factor was 2.36 points, while the degree of appraising it as a danger factor was 1.49 points. Comparing it with another study using the same instrument, in the research by Cha [22] targeting cancer patients, the appraisal as an opportunity (3.34 points) was higher than the appraisal as danger (2.21 points), which was similar to the results of this study. This could accord with the theory [20], in which once uncertainty is fixed as a normal part of life, uncertainty could be reappraised as an opportunity rather than a danger as a source that provides various opportunities inducing a state of positive emotion. 

In this study, self-efficacy averaged 6.04 points. This was lower than the 7.26 points shown in the research [37] targeting cancer survivors using the same instrument and also lower than the 6.68~6.69 points in the basic survey before intervention for the experimental study by Foster et al. [38]. Compared with the average age of research subjects in preceding studies, which was 54.2 years old [37] and 57.8 years old [38], the elderly patients with lung cancer in this study had lower self-efficacy in self-management than middle-aged cancer patients. The results of the preceding research show that self-efficacy could be reduced by aging, so there should be strategies for improving self-efficacy in the health care of elderly cancer patients. 

The quality of life of the subjects averaged 2.58 points. It was 2.50 in the research by Kim [39] which measured the quality of life by using the same instrument and 2.45 in the research by Lee and Kim [15], which is similar to the results of this study. In the sub-areas of quality of life, the physical state was average at 2.95 points, and the emotional state was average at 2.82 points, which was a bit good. However, the functional state was the lowest, at an average of 2.02 points. Such results are the same as the research by Jeong [14] and the research by Seo and Yi [40] reporting that the quality of life in a functional state was lower than the quality of life in a physical state and emotional state. To improve the quality of life of lung cancer patients, a nursing intervention considering their functional state would be necessary. 

In the quality of life according to the general characteristics of subjects, there were differences according to the degree of education and economic condition. This result accords with the research by Kim and Kim [41] and Park and Oh [42] reporting that the quality of life was high when the degree of education was higher, and the research by Bae [43] reported that the quality of life was high when the monthly income and economic condition were higher. Regarding the quality of life according to disease-related characteristics, the case of receiving immunotherapy or targeted therapy showed a higher quality of life than the case of receiving chemotherapy. In terms of the number of anticancer therapies, the first or second time showed higher quality of life than the third time or more, which supports the results of preceding research [44] showing that the quality of life was high when the number of anticancer therapies was less. Based on this, it would be necessary to have a nursing intervention that could improve self-management and quality of life, according to the number of anticancer therapies targeting elderly patients. 

The uncertainty and quality of life of the subjects showed an inverse correlation. When the uncertainty for lung cancer patients was higher, the quality of life got lower. Such results accord with the results of research by Lee [15] reporting the inverse correlation between uncertainty and quality of life in lung cancer patients, such as when uncertainty was higher, quality of life was lower. As a main psychological stress factor for cancer patients, uncertainty continues from the moment of diagnosis to the treatment process and even after treatment ends [45,46], which is a main factor that lowers the quality of life [47]. Thus, for managing and maintaining the quality of life, uncertainty needs to be managed. The appraisal of uncertainty, danger, and opportunity by subjects showed significant correlations with the quality of life. In other words, when the appraisal of uncertainty: the danger of subjects was lower, and when the appraisal of uncertainty: the opportunity was higher, the quality of life got higher. This accords with the results of research targeting prostate cancer patients by Nam et al. [23], who reported a significant inverse correlation between the quality of life and appraisal of uncertainty: danger, and a positive correlation between the quality of life and appraisal of uncertainty: opportunity. The uncertainty appraised as opportunity makes patients view their lives from a probabilistic or conditional perspective as the disease period is longer. They can positively cope with the disease by accepting the attributes of uncertainty as something natural as a result of probabilistic thinking, which could have effects on changing the health-related quality of life of patients [20]. Thus, if we reduce an uncertain situation in the treatment process and support subjects to have a positive appraisal of the uncertain situation based on the relation between quality of life and appraisal of uncertainty shown in this study, eventually the quality of life could be enhanced. 

The self-efficacy and quality of life of the subjects showed a positive correlation. When self-efficacy was higher, the quality of life got better. This accords with the results of research targeting breast cancer patients by Lee [27], showing a significant positive correlation between self-efficacy and quality of life. Self-efficacy, which is perceived confidence in the management of chronic disease, is an important factor for enjoying a healthy life and enhancing the quality of life [26]. Especially in the case of the elderly, the appraisal of self-ability could be different from adulthood due to the decline of self-ability, weakened physical/health functions, and loss of social relations with aging [48]. Thus, the quality of life should be increased by providing nursing interventions for maintaining self-efficacy.

According to hierarchical regression analysis, in the first stage, the quality of life was high in those with high school or higher education, and the quality of life was low in low economic conditions, when receiving anticancer therapy (chemotherapy), or when receiving chemotherapy three or more times. Their variables explained 36.5% of the subject’s quality of life. In the second stage, the subject’s quality of life decreased when receiving anticancer therapy (chemotherapy) or more than three times of anticancer therapy. Additionally, the quality-of-life score increased by 0.1 points every time the subject’s score perceived uncertainty evaluation as an opportunity increased by 1 point, and the quality of life decreased by 0.18 points every time the score perceived as a danger increased by 1 point. In addition, each time the self-efficacy score increased by 1 point, the quality of life increased by 0.16 points. Their variables explained 74.2% of the subject’s quality of life. 

In the first stage, among the general characteristics that affect the quality of life and disease-related characteristics, chemotherapy was found to have the highest influence. Symptoms such as fatigue, nausea, vomiting, loss of appetite, and weight loss resulting from chemotherapy make it difficult for patients to have confidence in their future [2]. As a result, the quality of life of patients undergoing chemotherapy decreases [49]. Therefore, interventions are needed to reduce side effects and improve the quality of life of elderly lung cancer patients receiving chemotherapy. The quality of life is higher among people with higher levels of education and greater economic status. These results are consistent with the findings of Kim [41], Park [42], and Bae [43], which show that higher education levels and higher economic status are associated with better quality of life. Given these results, it is important to actively focus on and manage the welfare of those with relatively low economic resources and education levels.

The study also revealed that the quality of life tends to decrease as the number of chemotherapy cycles increases. These findings are consistent with a recent study by Hyun and Kim [50], which showed that chemotherapy can be physically and mentally taxing for patients, and multiple cycles may require more intensive care and support to manage the adverse effects. Although chemotherapy is necessary for treating cancer, an increase in the number of cycles may indicate disease progression, which can be interpreted as a decrease in quality of life. Therefore, it is essential to provide individualized self-management education at the time of chemotherapy change to improve the quality of life of the subjects.

The most significant influencing factor in the second stage was self-efficacy, which affects quality of life. The study confirmed that higher self-efficacy is associated with a higher quality of life, which is consistent with other studies demonstrating the significant impact of self-efficacy on cancer patients’ quality of life [27,51]. According to Champion et al. [52], self-efficacy is associated with depression, anxiety, and fear of recurrence, which commonly occur after initial diagnosis and treatment. Thus, self-efficacy plays an important role in managing patients’ symptoms and improving their quality of life. Other studies by Papadopoulou et al. [53] also found a positive correlation between self-efficacy and depression, anxiety, and fear of recurrence, highlighting the importance of self-efficacy in managing patients’ symptoms and improving their overall quality of life. Therefore, nursing interventions that aim to increase self-efficacy are critical to improving the quality of life of cancer patients.

Next, the factors that were identified as affecting the quality of life were the appraisal of uncertainty as danger and the appraisal of uncertainty as opportunity. The appraisal of uncertainty has also been observed in studies of prostate cancer patients [23] and breast cancer patients [54] as major predictors of quality of life. Mishel [18] suggested that although the appraisal of uncertainty as a danger may have negative consequences, it can be transformed into an appraisal of uncertainty as an opportunity through the use of problem-oriented coping strategies or emotional coping strategies in order to reduce the negative impact. Therefore, assessing the uncertainty appraisal of lung cancer patients is crucial, and appropriate interventions should be implemented to help shift their appraisal of uncertainty from a focus on danger to one that emphasizes opportunity.

Meanwhile, in this study, it was confirmed that uncertainty does not affect quality of life. There was a difference from Kim’s [55] study, which evaluated the effect of uncertainty on the quality of life of cancer patients. This is because this study was for the elderly, and it is thought that there would have been a difference from Kim’s [55] study for the middle-aged. Therefore, in future studies, it will be necessary to compare the uncertainty of the elderly and the middle-aged and to understand the effect of uncertainty on the quality of life.

### Strengths and Limitations

This study is significant in the aspect of understanding the degree of uncertainty, appraisal of uncertainty, self-efficacy, and quality of life of elderly patients with lung cancer receiving anticancer therapy and also verifying the factors affecting the improvement of the quality of life of elderly patients with lung cancer. However, since this study included elderly patients with lung cancer from a single university hospital for convenience and did not include a sufficient number of subjects, generalizing the results of this study to elderly patients with lung cancer is limited. Furthermore, it needs to be careful when interpreting the results of this study, as there could be influences according to the treatment stage and symptoms following the types of anticancer drugs. 

## 5. Conclusions

As a result of influencing factors that affect the quality of life of elderly patients with lung cancer, self-efficacy, appraisal of uncertainty: danger, appraisal of uncertainty: opportunity, the number of anticancer therapy (three times or more), anticancer therapy (chemotherapy), economic condition (low), and education (graduation from high school or over) were significant influencing factors. Accordingly, in order to improve the quality of life of elderly patients with lung cancer, we analyze the degree to which uncertainty is evaluated as an opportunity or danger, reduce what is recognized as a danger, help them recognize it as an opportunity, and increase self-efficacy. In addition, intervention is needed in consideration of education, economic conditions, the type of anticancer therapy, and the number of anticancer therapy.

### Implications for the Profession and/or Patient Care/Clinical Practice

First, there should be expert interventions for improving the quality of life of elderly patients with lung cancer receiving anticancer therapy. It is highly possible for elderly patients with lung cancer to experience uncertainty and declining quality of life in the process of treating their cancer. Thus, to raise the quality of life of elderly patients with lung cancer, it is urgent to develop intervention programs for establishing effective management measures by considering the degree of education, economic condition, and types and several anticancer therapies. In particular, the type of treatment and the number of treatments require active expert intervention. Explanations to the patients and mutual cooperation between experts and patients will improve the quality of life.

In addition, there should be consideration of the appraisal of uncertainty: danger, opportunity, and self-efficacy. Furthermore, there should be research on the quality of life of cancer patients using Mishel’s theory of uncertainty.

## Figures and Tables

**Table 1 medicina-59-01051-t001:** Differences in the Quality of Life according to the General Characteristics and Disease-related Characteristics of Elderly Patients with Lung Cancer (*N* = 112).

Variables	Classification	*n*	%	Quality of Life		t/F	*p*-Value Scheffe Test
				Mean	SD		
Sex	Men	81	72.3	2.55	0.58	−0.87	0.387
	Wonen	31	27.7	2.66	0.65		
Age (years)	65–69	60	53.6	2.51	0.57	1.69	0.190
	70–79	40	35.7	2.72	0.62		
	80 and over	12	10.7	2.46	0.60		
Marital state	Married	79	70.5	2.62	0.54	1.15	0.254
	Unmarried, Divorced, etc.	33	29.5	2.48	0.72		
Main caregiver	Spouse	62	55.4	2.61	0.57	1.72	0.184
	Children	22	19.6	2.71	0.64		
	Caregiver	28	25.0	2.41	0.60		
Education	Graduated from middle school or lower	66	58.9	2.47	0.60	−2.48	0.015
	Graduated from high school or higher	46	41.1	2.73	0.56		
Breadwinner	Self, Pension	68	60.7	2.56	0.64	−0.53	0.601
	Spouse or Children	44	39.3	2.62	0.52		
Economic condition	Middle	61	54.5	2.76	0.58	3.69	<0.001
	Low	51	45.5	2.37	0.54		
Diagnosis	Small cell lung cancer	9	8.0	2.47	0.55	−0.57	0.571
	Non-small cell lung cancer	103	92.0	2.59	0.60		
Clinical stage	Stage I or II	9	8.0	2.49	0.60	−0.45	0.653
	Stage III or IV	103	92.0	2.59	0.60		
Types of anticancer therapy	Radiotherapy and chemotherapy ^a^	12	10.7	2.62	0.64	5.94	0.001
	Chemotherapy ^b^	28	25.0	2.20	0.55		b < c,d
	Immunotherapy ^c^	37	33.0	2.74	0.59		
	Targeted therapy ^d^	35	31.3	2.71	0.50		
The number of anticancer therapy	First ^a^	73	65.2	2.62	0.53	5.88	0.004
	Second ^b^	29	25.9	2.68	0.66		a,b > c
	Three times of more ^c^	10	8.9	2.00	0.56		

SD = standard deviation. “a,b,c,d”, “a,b,c”; Scheffe test.

**Table 2 medicina-59-01051-t002:** Degree of Variables Related to the Quality of Life of Subjects (*N* = 112).

Variables	Mean	SD	Actual Range
Uncertainty	2.62	0.49	1.48–3.64
Appraisal of uncertainty: danger	1.49	0.99	0–4.38
Appraisal of uncertainty: opportunity	2.36	1.03	0.57–5
Self-efficacy	6.04	1.53	1.90–9.20
Quality of life	2.58	0.95	1.07–3.72
Physical state	2.95	0.80	0.29–4
Social/family state	2.46	0.91	0–4
Emotional state	2.82	0.82	0.67–4
Functional state	2.02	0.89	0.29–4
Other state	2.65	0.75	0.43–4

SD = standard deviation.

**Table 3 medicina-59-01051-t003:** Correlations of Uncertainty, Appraisal of Uncertainty: Opportunity, Appraisal of Uncertainty: Danger, Self-efficacy, and Quality of Life (*N* = 112).

Variables	Uncertainty r (*p*)	Appraisal of Uncertainty: Danger r (*p*)	Appraisal of Uncertainty: Opportunity r (*p*)	Self-Efficacy r (*p*)	Quality of Life r (*p*)
Uncertainty	1				
Appraisal of uncertainty: danger	0.43 (<0.001)	1			
Appraisal of uncertainty: opportunity	−0.63 (<0.001)	−0.34 (<0.001)	1		
Self-efficacy	−0.46 (<0.001)	−0.38 (<0.001)	0.55 (<0.001)	1	
Quality of life	−0.56 (<0.001)	−0.61 (<0.001)	0.58 (<0.001)	0.70 (<0.001)	1

**Table 4 medicina-59-01051-t004:** Factors Affecting the Quality of Life of Elderly Patients with Lung Cancer.

Variables	Model 1					Model 2				
	B	SE	β	t	*p*	B	SE	β	t	*p*
Constants	2.82	0.11		24.76	<0.001	1.86	0.35		5.36	<0.001
Education (graduation from high school or over)	0.21	0.09	0.18	2.16	0.033	0.02	0.07	0.02	0.28	0.782
Economic condition (low)	−0.36	0.09	−0.30	−3.72	<0.001	−0.06	0.07	−0.05	−0.19	0.415
Anticancer therapy (radiotherapy + chemotherapy)	−0.19	0.17	−0.10	−1.12	0.264	−0.02	0.11	−0.01	−0.18	0.856
Anticancer therapy (chemotherapy)	−0.46	0.12	−0.34	−3.70	<0.001	−0.19	0.09	−0.14	−2.19	0.031
Anticancer therapy (targeted therapy)	0.06	0.12	0.04	0.52	0.602	0.12	0.08	0.09	1.45	0.151
The number of anticancer therapy (second)	0.04	0.12	0.03	0.32	0.751					
The number of anticancer therapy (three times or more)	−0.61	0.17	−0.29	−3.60	<0.001	−0.35	0.12	−0.17	−2.84	0.006
The number of anticancer therapy (first)						0.03	0.08	0.02	0.39	0.698
Uncertainty						−0.07	0.09	−0.06	−0.76	0.448
Appraisal of uncertainty: opportunity						0.10	0.04	0.18	2.40	0.018
Appraisal of uncertainty: danger						−0.18	0.04	−0.29	−4.73	<0.001
Self-efficacy						0.16	0.03	0.41	5.97	<0.001
R^2^	0.365					0.742				
Adjusted R^2^	0.322					0.714				
△ Adjusted R^2^ (*p*)						0.392 (<0.001)				
F *(p)*	8.52 (<0.001)					26.17 (<0.001)				

B = unstandardized regression coefficient; SE = standard error; β = standardized regression coefficient, References: Education (graduation from middle school or under); economic condition (low); chemotherapy (radiotherapy + chemotherapy); the number of chemotherapy (first); the number of chemotherapy (second).

## Data Availability

Our data are readily available upon reasonable request.

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
