# Peer review of "Effects of Uncertainty, Appraisal of Uncertainty, and Self-Efficacy on the Quality of Life of Elderly Patients with Lung Cancer Receiving Chemotherapy: Based on Mishel’s Theory of Uncertainty"

_medicina, 2023, doi:10.3390/medicina59061051_

Round 1

Reviewer 1 Report

  • Where did the patients perform the self-completion?
  • The wording of results should be improved. 
  • Conclusions should be improved. It needs to be careful when interpreting the results of this study as there could be influences according to the treatment stage and symptoms following the types of anticancer drug.
  • The study is original although it does not provide solid conclusions that contribute knowledge to science. The conclusions or interpretations of the results should be improved.

Author Response

Reviewer 1

Thank you very much for your careful review.

I revised it carefully as you pointed out.

 Where did the patients perform the self-completion?

Added

I met the subjects in the outpatient counseling room of the Department of Hematology at Chungbuk National University Hospital and collected data.

 The wording of results should be improved. 

Changed and improved the results

Reduce general characteristics to focus on important areas and modify the interpretation of correlations
The factors influencing the quality of life are described according to the magnitude of the importance of the variable.

 Conclusions should be improved. It needs to be careful when interpreting the results of this study as there could be influences according to the treatment stage and symptoms following the types of anticancer drug.

Changed conclusion

This study aimed to understand the effects of uncertainty, appraisal of uncertainty, and self-efficacy on the quality of life targeting elderly patients with lung cancer. As a result of influencing factors that affect the quality of life of elderly patients with lung cancer, education, economic condition, the type of chemotherapy, the number of chemotherapy, and appraisal of uncertainty: opportunity, appraisal of uncertainty: danger and self-efficacy were significant effecting factors. Accordingly, in order to improve the quality of life of elderly patients with lung cancer, we analyze the degree to which uncertainty is evaluated as an opportunity or danger, reduce what is recognized as a danger, and help them to recognize it as an opportunity, and increase self-efficacy. In addition, intervention is needed in consideration of education, economic conditions, and the type and number of chemotherapy.

Implications for the profession and/or patient care/clinical practice are as follows. First, there should be experts’ interventions for improving the quality of life of elderly patients with lung cancer receiving anticancer therapy. It is highly possible for elderly patients with lung cancer to experience uncertainty and declined quality of life in the process of treating their cancer. Thus, to raise the quality of life of elderly patients with lung cancer, it is urgent to develop intervention programs for establishing the effective management measures by considering the degree of education, economic condition, and types and number of anticancer therapy, In particular, the type of treatment and the number of treatments require active expert intervention. Explanation to the patients and mutual cooperation between experts and patients will improve the quality of life.

And also considering the appraisal of uncertainty: danger & opportunity and self-efficacy. And there should be researches on the quality of life of cancer patients using Mishel’s Theory of Uncertainty.

 The study is original although it does not provide solid conclusions that contribute knowledge to science. The conclusions or interpretations of the results should be improved.

Thank you so much.

The conclusions or interpretations of the results were improved.

4.1. Strengths and limitations

This study is significant in the aspect of understanding the degree of uncertainty, appraisal of uncertainty, self-efficacy, and quality of life of elderly patients with lung cancer receiving anticancer therapy, and also verifying the factors affecting the improvement of the quality of life of elderly patients with lung cancer. In particular, Michelle's theory of uncertainty and uncertainty evaluation was used to derive results and applied to chronic diseases such as cancer, resulting in an expansion of research and theory. However, since this study included elderly patients with lung cancer from a single university hospital for convenience and did not include a sufficient number of subjects, generalizing the results of this study to elderly patients with lung cancer is limited. Also, it needs to be careful when interpreting the results of this study as there could be influences according to the treatment stage and symptoms following the types of anticancer drug.

5. Conclusions

This study aimed to understand the effects of uncertainty, appraisal of uncertainty, and self-efficacy on the quality of life targeting elderly patients with lung cancer. As a result of influencing factors that affect the quality of life of elderly patients with lung cancer, self-efficacy, appraisal of uncertainty: danger, appraisal of uncertainty: opportunity, the number of anticancer therapy, anticancer therapy (chemotherapy), economic condition (low), and education (graduation from high school or over) were significant effecting factors. Accordingly, in order to improve the quality of life of elderly patients with lung cancer, we analyze the degree to which uncertainty is evaluated as an opportunity or danger, reduce what is recognized as a danger, and help them to recognize it as an opportunity, and increase self-efficacy. In addition, intervention is needed in consideration of education, economic conditions, and the type and number of chemotherapy.

Implications for the profession and/or patient care/clinical practice are as follows. First, there should be experts’ interventions for improving the quality of life of elderly patients with lung cancer receiving anticancer therapy. It is highly possible for elderly patients with lung cancer to experience uncertainty and declined quality of life in the process of treating their cancer. Thus, to raise the quality of life of elderly patients with lung cancer, it is urgent to develop intervention programs for establishing the effective management measures by considering the degree of education, economic condition, and types and number of anticancer therapy, In particular, the type of treatment and the number of treatments require active expert intervention. Explanation to the patients and mutual cooperation between experts and patients will improve the quality of life.

And also considering the appraisal of uncertainty: danger & opportunity and self-efficacy. And there should be researches on the quality of life of cancer patients using Mishel’s Theory of Uncertainty.

Reviewer 2 Report

Thanks for the authors,  regarding the comments of your paper, please see the below file.

Author Response

Reviewer 2

Thank you very much for your careful review.

I revised it carefully as you pointed out.

1. Line 6, 2* should be 2,*

revised

2,*

2. Line 7, Gongju 32588; should be Gongju 32588,

revised

32588,

3. Line 17, C university, Please specify which university

revised

Chungbuk National University Hospital.

4. Line 19, The quality of life participants, should be "The quality of life of participants"

revised

The quality of life of participants

5. Line 20, dander should be danger

revised

danger

6. Please explain how the range of r values is defined as highly correlated

revised

moderate

r=0.2~0.4> low correlation

r=0.4~0.7-> moderate correlation

r=0.7~0.9-> high correlation

r> 0.9-> very high correlation

7. Line 22,25 stage 1/2 should stage 1/2...

revised

stage 1

stage 2

Introduction

8. I suggest the contents of Paragraphs 1-4 be more concise, go straight to the topic of quality of life among lung cancer patients, and the importance of studying the quality of life issues in lung cancer patients.

It simplifies the introduction and adds the definition and  important of quality of life, and Michelle's uncertainty

Introduction

In Korea, the elderly population aged 65 or older was 14.9% in 2019, which marked the entry into an aged society. It is forecasted to enter a super-aged society by reaching 25.5% in 2023 [1]. Furthermore, the incidence rate of cancer in people aged 65 or older was 47.8% of all patients, mainly due to the natural increase in cancer incidence following an aging population, making it a significant cause for the recently increased number of cancer patients [2]. Especially, lung cancer is the most prevalent among the elderly population aged 65 or older [1,2].

Meanwhile, thanks to the recent expansion of health medical examination and the development of treatment methods, there has been an increase in the early detection rate, and the five-year relative survival rate of lung cancer patients has also improved [1]. This means that the period of struggling against cancer is extended after diagnosis [3]. Furthermore, with the increased survival rate of cancer patients, the medical team's interest is gradually expanding from reducing the tumor size to improving the quality of life of patients [4].

In the case of lung cancer patients, the question of "what kind of life he/she should live?" is becoming increasingly important [5], making the quality of life a treatment goal second only to survival rate. In fact, recent studies have revealed that the quality of life is a strong predictive factor of survival period, highlighting its growing importance [6,7]. For instance, in a study conducted by Yun et al. [8], the risk of death for lung cancer patients with reduced health-related quality of life after surgery was more than twice as high as that of patients with no change in quality of life. This underscores the importance of managing and improving the quality of life of lung cancer patients.

Moreover, the symptoms that lung cancer patients experience can have a negative impact on their physical, psychological, social, and spiritual well-being [9]. Inadequate management of these symptoms may lead to psychological distress and cause a decline in physical function and quality of life [10,11]. Therefore, it is crucial to properly manage the symptoms and improve the quality of life of lung cancer patients.

The quality of life is a state of well-being that is subjectively appraised based on the general and overall situation or life experience in multidimensional areas such as physical, psychological, social, and functional areas [12]. For cancer patients, the quality of life is influenced by factors such as clinical stage, age, and performance status at the time of diagnosis [13]. Also, greater uncertainty about their condition can destabilize a patient's emotional, physical, and social state, leading to a decline in their overall quality of life [14-16]. In another study by Kim [17] comparing uncertainty levels between adults and elderly patients with lung cancer, the elderly had higher levels of uncertainty about the disease due to sociodemographic and disease-related characteristics.

Uncertainty refers to a lack of understanding of one's disease, related treatments, post-treatment prognosis, and the inability to judge disease-related situations. If uncertainty persists throughout the course of the disease's progression, it can be considered threatening as it delays the formation of cognitive structures and limits the ability to properly assess an individual's situation [18]. According to Michelle [19], uncertainty in disease-related situations arises when decision makers cannot accurately evaluate goals or predict results. Therefore, it is important for experts to help reduce uncertainty by assisting patients in recognizing disease conditions and situations. Furthermore, Michelle [20] notes that individuals in a state of chronic uncertainty shift from trying to avoid it to accepting it as a new perspective on life as a part of reality.

Because uncertainty in the early stage of disease becomes a changeable and destructive element, a patient appraises it as ‘danger’. Once the uncertainty is continued, a certain order is built and the uncertainty is appraised positively. In this case, it could be regarded as an ‘opportunity’ of life [21], so the appraisal of uncertainty is also important. In other words, if uncertainty is appraised as danger by an individual, it can cause negative results. On the other hand, if it is appraised as opportunity, the ability to cope with the disease is improved and a new view of life could be established [22]. Thus, in order for patients to be able to positively cope with disease process, it is important to help them integrate uncertainty that was appraised as danger as a part of their lives, and change it to a positive opportunity [23].

Also, self-efficacy is a dynamic process of appraising one’s own ability to perform actions necessary for coping with and adjusting to a potentially-threatening event [24]. Because the expectation in which a certain act brings about a certain result, and confidence in the successful performance of a certain act have huge effects on actions [24], cancer patients can continuously maintain their disease control by having self-efficacy [25]. Self-efficacy which is confidence in managing chronic disease is an important factor for enjoying healthy life and enhancing the quality of life [26]. For this reason, it plays an important role in cancer patients who should endlessly practice self-management [27,28]. In a study conducted by Ko [29] on non-small cell lung cancer patients, higher levels of self-efficacy were associated with better quality of life.

To cancer patients, the quality of life is such an important goal as much as survival, so there have been many researches on it, but it is hard to find researches targeting elderly patients with lung cancer. There have been many studies conducted on the importance of quality of life for cancer patients, as it is just as crucial as their survival. However, it has been difficult to find studies specifically focused on elderly patients with lung cancer. Therefore, the purpose of this study aims to provide basic data for the development of interventions for enhancing the quality of life in elderly patients with lung cancer, by understanding relations of uncertainty, appraisal of uncertainty: danger and opportunity, self-efficacy, and quality of life targeting elderly patients with lung cancer receiving anticancer therapy, and also analyzing the factors affecting the quality of life based on Mishel’s Theory of Uncertainty.

9. Line 65,74 be led to or led to? passive voice or active voice? Please check.

revised

Line 65: Moreover, the symptoms that lung cancer patients experience can have a negative impact on their physical, psychological, social, and spiritual well-being [9]. Inadequate management of these symptoms may lead to psychological distress and cause a decline in physical function and quality of life [10,11]. Therefore, it is crucial to properly manage the symptoms and improve the quality of life of lung cancer patients.

Line 74: Also, greater uncertainty about their condition can destabilize a patient's emotional, physical, and social state, leading to a decline in their overall quality of life [14-16].

10. Line 79, the citation for the definition of uncertainty needs to be added.

added

Uncertainty refers to a lack of understanding of one's disease, related treatments, post-treatment prognosis, and the inability to judge disease-related situations. If uncertainty persists throughout the course of the disease's progression, it can be considered threatening as it delays the formation of cognitive structures and limits the ability to properly assess an individual's situation [18].

11. Line 94, the content of Mishel’s Theory of Uncertainty is not sufficiently described, Please supplement appropriately.

added

According to Michelle [19], uncertainty in disease-related situations arises when decision makers cannot accurately evaluate goals or predict results. Therefore, it is important for experts to help reduce uncertainty by assisting patients in recognizing disease conditions and situations. Furthermore, Michelle [20] notes that individuals in a state of chronic uncertainty shift from trying to avoid it to accepting it as a new perspective on life as a part of reality.

12. Line 101, spelling check is needed, dancer?

revised

danger

13. Line136-137, eight independent variables? I think the sample size should be calculated based on the initial number of independent variables, please check. Your sample size 112 is a bit small. The initial number of independent variables is 11+4=15.

We learned that specific sample sizes are calculated including independent variables and variables that are expected to affect general characteristics.
Therefore, it does not include all variables.
Currently, the subjects of the study are elderly lung cancer patients, and it is very difficult to obtain the subjects, and it is very difficult to complete the questionnaire, so we calculated the minimum number of people based on convenience sampling. Please understand this. I'll try to get more people in the next time I do research.
The number of people has already been confirmed through IRB, so it is difficult to adjust the number of people. I will include this part in the limitations of the study.

14. Line 153, 13 should be thirteen

revised

thirteen

15. Line 198, C university, Please specify which university

revised

Chungbuk National University Hospital

Result

16. Line 224-236, Do not repeat the contents in the text and table too much.

Deleted a lot, and only described the least important parts

The general characteristics and disease-related characteristics of subjects are as Table 1. In the general characteristics, the number of elderly patients with lung cancer was 112. The average age was 70.98±5.65Y: the range of age was 65~87Y. The most responses were obtained in married (N=79, 70.5%) for marital status, spouse (N=62, 55.4%) for main caregiver, and oneself or pension (N=68, 60.7%) for breadwinner. In the disease-related characteristics, the most responses were obtained in non-small cell lung cancer (N=103, 92.0%) for diagnosis, and 3-4 stage (N=103, 92.0%) for clinical stage. The types of anticancer therapy included immunotherapy (N=37, 33.0%), targeted therapy (N=35, 31.3%), chemotherapy (N=28, 25.0%), and radiotherapy and chemotherapy (N=12, 10.7%). In the number of receiving anticancer therapy, 73 people (65.2%) received it once.

17. Table 1, stage 1-2/3-4, Please use Roman numerals

revised

Stage Ι or Π

Stage Ш or ΙШ

18. Line 247-255, 275-278, The conclusion should be interpreted with caution. It can only indicate the correlations between variables.

revised

Line 247-253 is a description of the results of comparative research, so it was explained as a comparison between groups.
line 275-278 is modified to describe correlation.

In other words, the higher the degree of uncertainty and the higher the appraisal of uncertainty as a danger, the lower the quality of life of the participants. In addition, the higher the degree of self-efficacy and the higher the appraisal of uncertainty as an opportunity, the higher the quality of life of the participants.

19. Table 4, please check.

corrected the wrong part.

Discussion

20. Line 425, influence factors should be influencing factors

revised

influencing factors

21. I suggest that the discussion should focus on influencing factors, e.g. results table4, B or β values should also be explained in the discussion section.

revised, and added

According to hierarchical regression analysis, in the first stage, anticancer therapy (chemotherapy) (β=-0.34), economic condition (low) (β=-0.30), the number of anticancer therapy (three times or more) (β=-0.29), and education (graduation from high school or over) (β=0.18), were expressed in the order of high influence among general characteristics and disease-related characteristics that affect the quality of life (F=8.52, p<0.001). Their variables explained 36.5% of the subject's quality of life. In the second stage, self-efficacy (β=0.41), appraisal of uncertainty: danger (β=-0.29), appraisal of uncertainty: opportunity (β=0.18), the number of anticancer therapy (three times or more) (β=-0.17), and anticancer therapy (chemotherapy) (β=-0.14) were expressed in the order of high influence that affect the quality of life. Their variables explained 74.2% of the subject's quality of life (F=26.17, p<0.001).

In the first stage, among the general characteristics that affect quality of life and disease-related characteristics, chemotherapy was found to have the highest influence. Symptoms such as fatigue, nausea, vomiting, loss of appetite, and weight loss resulting from chemotherapy make it difficult for patients to have confidence in their future [2]. As a result, the quality of life of patients undergoing chemotherapy decreases [50]. Therefore, interventions are needed to reduce side effects and improve the quality of life of elderly lung cancer patients receiving chemotherapy. The quality of life is higher among people with higher levels of education and greater economic status. These results are consistent with the findings of Kim [41], Park [42], and Bae [43], which show that higher education levels and higher economic status are associated with better quality of life. Given these results, it is important to actively focus on and manage the welfare of those with relatively low economic resources and education levels.

The study also revealed that the quality of life tends to decrease as the number of chemotherapy cycles increases. These findings are consistent with a recent study by Hyun and Kim [51], which showed that chemotherapy can be physically and mentally taxing for patients, and multiple cycles may require more intensive care and support to manage the adverse effects. Although chemotherapy is necessary for treating cancer, an increase in the number of cycles may indicate disease progression, which can be interpreted as a decrease in quality of life. Therefore, it is essential to provide individualized self-management education at the time of chemotherapy change to improve the quality of life of the subjects.

The most significant influencing factor in the second stage was self-efficacy, which affects the quality of life. The study confirmed that higher self-efficacy is associated with a higher quality of life, which is consistent with other studies demonstrating the significant impact of self-efficacy on cancer patients' quality of life [48,52]. According to Champion et al. [53], self-efficacy is associated with depression, anxiety, and fear of recurrence, which commonly occur after initial diagnosis and treatment. Thus, self-efficacy plays an important role in managing patients' symptoms and improving their quality of life. Other studies by Papadopoulou et al. [54] also found a positive correlation between self-efficacy and depression, anxiety, and fear of recurrence, highlighting the importance of self-efficacy in managing patients' symptoms and improving overall quality of life. Therefore, nursing interventions that aim to increase self-efficacy are critical to improving the quality of life of cancer patients.

Next, the factors that were identified as affecting quality of life were appraisal of uncertainty as danger and appraisal of uncertainty as opportunity. The appraisal of uncertainty has also been observed in studies of prostate cancer patients [23] and breast cancer patients [55] as major predictors of quality of life. Mishel [18] suggested that although the appraisal of uncertainty as danger may have negative consequences, it can be transformed into an appraisal of uncertainty as opportunity through the use of problem-oriented coping strategies or emotional coping strategies in order to reduce the negative impact. Therefore, assessing the uncertainty appraisal of lung cancer patients is crucial, and appropriate interventions should be implemented to help shift their appraisal of uncertainty from a focus on danger to one that emphasizes opportunity.

Meanwhile, in this study, it was confirmed that uncertainty does not affect quality of life. There was a difference from Kim [56] study, which evaluated the effect of uncertainty on the quality of life of cancer patients. This is because this study was for the elderly, and it is thought that there would have been a difference from Kim [56] study for the middle-aged. Therefore, in future studies, it is necessary to compare the uncertainty of the elderly and the middle-aged, and to understand the effect of uncertainty on the quality of life.

22. The discussion section did not provide a good description regarding the guiding significance of Mishel’s Theory of Uncertainty for this study. Please add these contents.

Added and revised

Next, the factors that were identified as affecting quality of life were appraisal of uncertainty as danger and appraisal of uncertainty as opportunity. The appraisal of uncertainty has also been observed in studies of prostate cancer patients [23] and breast cancer patients [55] as major predictors of quality of life. Mishel [18] suggested that although the appraisal of uncertainty as danger may have negative consequences, it can be transformed into an appraisal of uncertainty as opportunity through the use of problem-oriented coping strategies or emotional coping strategies in order to reduce the negative impact. Therefore, assessing the uncertainty appraisal of lung cancer patients is crucial, and appropriate interventions should be implemented to help shift their appraisal of uncertainty from a focus on danger to one that emphasizes opportunity.

23. Line 435-442, The authors can consider adding a title “4.1 strengths and limitations”

Added title

4.1. Strengths and limitations

This study is significant in the aspect of understanding the degree of uncertainty, appraisal of uncertainty, self-efficacy, and quality of life of elderly patients with lung cancer receiving anticancer therapy, and also verifying the factors affecting the improvement of the quality of life of elderly patients with lung cancer. However, since this study included elderly patients with lung cancer from a single university hospital for convenience and did not include a sufficient number of subjects, generalizing the results of this study to elderly patients with lung cancer is limited. Also, it needs to be careful when interpreting the results of this study as there could be influences according to the treatment stage and symptoms following the types of anticancer drug.

Conclusions

24. Please keep the content concise, it's too long

Deleted sentences. Rewrote them.

This study aimed to understand the effects of uncertainty, appraisal of uncertainty, and self-efficacy on the quality of life targeting elderly patients with lung cancer. As a result of influencing factors that affect the quality of life of elderly patients with lung cancer, education, economic condition, the type of chemotherapy, the number of chemotherapy, and appraisal of uncertainty: opportunity, appraisal of uncertainty: danger and self-efficacy were significant effecting factors. Accordingly, in order to improve the quality of life of elderly patients with lung cancer, we analyze the degree to which uncertainty is evaluated as an opportunity or danger, reduce what is recognized as a danger, and help them to recognize it as an opportunity, and increase self-efficacy. In addition, intervention is needed in consideration of education, economic conditions, and the type and number of chemotherapy.

25. Line 459-468, The authors can consider adding a title “Implications for the profession and/or patient care/clinical practice...”

Added title

Implications for the profession and/or patient care/clinical practice was as follows.

Implications for the profession and/or patient care/clinical practice are as follows. First, there should be experts’ interventions for improving the quality of life of elderly patients with lung cancer receiving anticancer therapy. It is highly possible for elderly patients with lung cancer to experience uncertainty and declined quality of life in the process of treating their cancer. Thus, to raise the quality of life of elderly patients with lung cancer, it is urgent to develop intervention programs for establishing the effective management measures by considering the degree of education, economic condition, and types and number of anticancer therapy, In particular, the type of treatment and the number of treatments require active expert intervention. Explanation to the patients and mutual cooperation between experts and patients will improve the quality of life.

And also considering the appraisal of uncertainty: danger & opportunity and self-efficacy. And there should be researches on the quality of life of cancer patients using Mishel’s Theory of Uncertainty.

26. Finally, please check the format of each

checked the format of each

Round 2

Reviewer 2 Report

Thanks for the authors, they made lots of revisions. But the revised manuscript is very difficult to review,a large number of red revisions made my eyes very uncomfortable. Any revisions to your paper should be  marked up using the "track changes" function, such that  any changes can be easily viewed by the editors and reviewers. You shoud better provide a cover letter (not WORD type of a revised manuscript) instead of pasting a large amount of text words into the box. 

1. The introduction is too long. Some paragraphs could be more concise. e.g., paragraph 1.,  only "lung cancer is the most prevalent among the elderly population aged 65 or older".

2. Line 128-132,I suggest deleting this content "The number of samples required for this study was calculated by using the G-power 3.1.9 Program. In case of conducting the multiple regression analy-sis by setting up as significance level (α) .05, effect size .15, test power (1-β) 0.80, and eight independent variables, the minimum number of required samples was calculated as 109 people. Thus, total 114 subjects were calculated by considering the drop-out rate (5%).", Because there are some doubts about these contents. e.g., the drop-out rate (5%) or 10%-20%? and the number of independent variables...

3. Line 224 and table 1, 1-2-3-4 stage should be  ⅠⅡ  Ⅲ   Ⅳ

4. please check Table 4 .

In model 2, regarding the number of anticancer therapy,   first  or  second  is reference? there  are no data in “The number of anticancer therapy (second)“ box.

5. Line 405-415,Not to ask you to repeat the results, but to explain what the B or β value represents.   e.g.,"for every 1 point increase in self‑efficacy score, the patient’s QOL score would increase by 0.16 points."

6. Lines 487-500,  This paragraph should be presented as a separate section.  

   6. Implications for the profession and/or patient care/clinical practice

7.  line 476-177,“This study aimed to understand the effects of uncertainty, appraisal of uncertainty, and self-efficacy on the quality of life targeting elderly patients with lung cancer.” should be deleted. The conclusion is too long.

8. line 512,  Should end with a “

9. check the format of each ref.      e.g.,  [6]  and others.  Please maintain a consistent format.

10. Why did the authors build two models instead of directly buliding model 2?   Can't you directly include 8 independent variables into the multivariate analysis? 

11. Please check each data in tables 1-4 again, ensure that they are all correct.

Author Response

Reviewer 2

Thank you for reviewing to be a good paper. I revised it hard as the reviewer pointed out.

1. The introduction is too long. Some paragraphs could be more concise. e.g., paragraph 1.,  only "lung cancer is the most prevalent among the elderly population aged 65 or older".

1. 서론이 너무 깁니다. 일부 단락은 더 간결할 수 있습니다. 예를 들어, 단락 1. "폐암은 65세 이상 노인 인구 중에서 가장 보편적입니다."

Apply “Track changes” function.

In introduction, deleted sentence, and shortening

In Korea, the elderly population aged 65 or older was 14.9% in 2019 [1]. Furthermore, the incidence rate of cancer in people aged 65 or older was 47.8% of all patients, mainly due to the natural increase in cancer incidence following an aging population, making it a significant cause for the recently increased number of cancer patients [2].

Meanwhile, thanks to the recent expansion of health medical examination and the development of treatment methods, the five-year relative survival rate of lung cancer patients has also improved [1]. This means that the period of struggling against cancer is extended after diagnosis [3]. Furthermore, with the increased survival rate of cancer patients, the medical team's interest is gradually expanding from reducing the tumor size to improving the quality of life of patients [4]. In the case of lung cancer patients, the question of "what kind of life he/she should live?" is becoming increasingly important [5], making the quality of life a treatment goal second only to survival rate.

2. Line 128-132,I suggest deleting this content "The number of samples required for this study was calculated by using the G-power 3.1.9 Program. In case of conducting the multiple regression analy-sis by setting up as significance level (α) .05, effect size .15, test power (1-β) 0.80, and eight independent variables, the minimum number of required samples was calculated as 109 people. Thus, total 114 subjects were calculated by considering the drop-out rate (5%).", Because there are some doubts about these contents. e.g., the drop-out rate (5%) or 10%-20%? and the number of independent variables...

Line 128-132, deleted

The number of samples required ~~~ by considering the drop-out rate (5%).

3. Line 224 and table 1, 1-2-3-4 stage should be  ⅠⅡ  Ⅲ   Ⅳ

Changed

stage Ⅲ or Ⅳ

4. please check Table 4 .

In model 2, regarding the number of anticancer therapy,   first or  second  is reference? there are no data in “The number of anticancer therapy (second)“ box.

Add the reference below the table 4.

In Model 1, chemotherapy (first) is a reference, and in Model 2, chemotherapy (second) is a reference. Thank you.

5. Line 405-415,Not to ask you to repeat the results, but to explain what the B or β value represents.   e.g.,"for every 1 point increase in self‑efficacy score, the patient’s QOL score would increase by 0.16 points."

Revised

According to hierarchical regression analysis, in the first stage, the quality of life was high in those with high school or higher education, and the quality of life was low in low economic conditions, when receiving anticancer therapy (chemotherapy), or when receiving chemotherapy three or more times. Their variables explained 36.5% of the subject's quality of life. In the second stage, the subject's quality of life decreased when receiving anticancer therapy (cancer therapy) or more than three times of chemotherapy. And the quality of life score increased by 0.1 points every time the subject's score perceived uncertainty evaluation as an opportunity increased by 1 point, and the quality of life decreased by 0.18 points every time the score perceived as a danger increased by 1 point. In addition, each time the self-efficacy score increased by 1 point, the quality of life increased by 0.16 points. Their variables explained 74.2% of the subject's quality of life.

6. Lines 487-500,  This paragraph should be presented as a separate section. 

Implications for the profession and/or patient care/clinical practice

Add the 5.1 title

5.1. Implications for the profession and/or patient care/clinical practice

7.  line 476-177,“This study aimed to understand the effects of uncertainty, appraisal of uncertainty, and self-efficacy on the quality of life targeting elderly patients with lung cancer.” should be deleted. The conclusion is too long.

Deleted the sentence.

8. line 512,  Should end with a “ .  ”

Added .(period)

9. check the format of each ref.      e.g., [6] and others.  Please maintain a consistent format.

Changed reference 6, 9. And checked all

Lee, J.L. Quality of life Non-small Cell Lung Cancer Patients: A Structural Equation Model Approach [Doctoral dissertation]. Seoul National University, Seoul, Republic of Korea, 2013.

Fan G.; Filipczak L.; Chow E. Symptom Clusters in Cancer Patients: A Review of the Literature. Curr. Oncol. 2007, 14, 173-9. http://dx.doi.org/10.34747/co.2007.145

10. Why did the authors build two models instead of directly buliding model 2?   Can't you directly include 8 independent variables into the multivariate analysis?

The researchers analyzed the model by dividing it into two. The reason was to first understand the influence on the quality of life with the general characteristics and disease-related characteristics of the subject. After that, we wanted to analyze how much influence it shows on the quality of life when we include interventionable variables. Furthermore, we tried to use it as basic data for the intervention program in the future.

11. Please check each data in tables 1-4 again, ensure that they are all correct.

Rechecked table 1-4
